



# Joint Occurrence of Heatwaves and Ozone Pollution and Increased Health Risks in Beijing, China: Roles of Synoptic Weather Pattern and Urbanization

Lian Zong[1],Yuanjian Yang[1,*], Haiyun Xia[1,*], Meng Gao[2], Zhaobin Sun[3], Zuofang Zheng[3], Xianxiang Li[4], Guicai Ning[5], Yubin Li[1], Simone Lolli[6]

[1]Collaborative Innovation Centre on Forecast and Evaluation of Meteorological Disasters, Key Laboratory for Aerosol-Cloud-Precipitation of China Meteorological Administration, School of Atmospheric Physics, Nanjing University of Information Science & Technology, Nanjing, China.
[2]Department of Geography, Hong Kong Baptist University, Hong Kong, China.
[3]Institute of Urban Meteorology, China Meteorological Administration, Beijing, China.
[4]School of Atmospheric Sciences, Sun Yat-Sen University, Guangzhou, China.
[5]Department of Land Surveying and Geo-Informatics, The Hong Kong Polytechnic University, Hong Kong, China.
[6]CNR-IMAA, Contrada S. Loja, 85050 Tito Scalo (PZ), Italy

*Correspondence to*: Prof. Yuanjian Yang (yyj1985@nuist.edu.cn) and Prof. Haiyun Xia (hsia@ustc.edu.cn)

**Abstract.** Heatwaves (HWs) paired with higher ozone ($O_3$) concentration at surface level pose a serious threat to human health. Their combined modulation of synoptic patterns and urbanization remains unclear. By using five years of summertime temperature and $O_3$ concentrations observation in Beijing, this study explored potential drivers of compound HWs and $O_3$ pollution events. Three unfavourable synoptic weather patterns were identified to dominate the compound HWs and $O_3$ pollution events. The weather patterns contributing to enhance those conditions are characterized by sinking air motion, low boundary layer height, and hot temperatures. Under the synergistic stress of HWs and $O_3$ pollution, the public mortality risk increased by approximately 12.59% (95% confidence interval: 4.66%, 21.42%). Relative to rural areas, urbanization caused higher risks for HWs, but lower risks for $O_3$ over urban areas. In general, unfavourable synoptic patterns and urbanization can enhance the compound risk of events in Beijing by 45.46% and 8.08%, respectively. Our findings provide robust evidence and implications for forecasting compound heatwaves and $O_3$ pollution event and its health risks in Beijing or in other urban areas all over the word having high concentrations of $O_3$ and high-density populations.

**Key words:** Heatwaves, ozone pollution, compound health risks, synoptic weather pattern, urbanization

## 1 Introduction

Climate warming and rapid urbanization have led to an increase in the frequency and duration of extreme high-temperature episodes (Lehner et al., 2018; Meehl & Tebaldi, 2004; Wang et al., 2021b; Yang et al., 2017; Li, 2020), to the point that they are recognized as one of the most serious types of meteorological disaster worldwide. Prolonged extreme high-temperature exposure can induce an increase in the morbidity and mortality due to cardiovascular and respiratory diseases, posing a serious



threat to human health (Patz et al., 2005; Xu et al., 2016). Together with the rapid development of economic globalization and urbanization, human activities and the changes in the urban underlying surface have enhanced urban hyperthermia and frequent air pollution issues (Chew, et al., 2021; Li et al., 2016; Luo & Lau, 2018, 2019; Meehl et al., 2007; Rastogi, 2020; Wang et al.,

2007; Yang et al., 2020; Zheng et al., 2020). Heat waves (HWs) paired with the urban heat island (UHI) effect exposes urban residents to more sustained extreme high temperatures (Chew et al., 2021; Jiang et al., 2019; Tan et al., 2010; Wang et al., 2017; Zong et al., 2021b). The strong solar radiation and high temperatures in summer have accelerated the photochemical reaction, which is a proxy to the production and accumulation of $O_3$ (Herring et al., 2019; Shu et al., 2016; Zanis et al., 2000, 2011). In particular, $O_3$ became the main pollutant in summer (Fan et al., 2020; Gao et al., 2020b; Li et al., 2019; Wang et al.,

2019; Saikawa et al., 2017), and thus residents may suffer from dual health risks brought about by high temperatures and $O_3$ exposure in summer.

Accumulating epidemiological evidence shows that the coefficient of the $O_3$ concentration–response relationship for mortality in summer is higher (Atkinson et al., 2012; Pattenden et al., 2010; Pope et al., 2016; Zhong et al., 2019), suggesting that the health effects and mortality related to $O_3$ pollution are exacerbated by hot temperatures. Therefore, the significant increase in

$O_3$ concentrations during summertime is also greatly responsible for the increase in excess mortality; high temperatures and $O_3$ have a joint impact on public health (Hertig et al., 2020; Katsouyanni et al., 1993; Lelieveld et al., 2014; Pattenden et al., 2010). Numerous studies have been devoted to the individual impacts of a single extreme high-temperature or air pollution event on human health (Ma et al., 2015; Ning et al., 2020; Wang et al., 2020; Wong et al., 2013; Xu et al., 2016). However, with the co-occurrence of extreme HW and $O_3$ pollution events in summer becoming more frequent, it is imperative to reveal

the underlying mechanisms of extreme HW–$O_3$ compound events and improve the level of risk assessment related to extreme events in urban areas (Sartor et al. 1995; Hertig et al., 2020).

Beijing, the capital of China, is the second largest city in the country, with a permanent population of 21.89 million, making it one of the fastest developing metropolises in recent decades. Rapid urbanization and urban expansion have induced changes in land-use types, which in turn have changed the energy budget within the city boundary layer (Dou et al., 2015; Li et al.,

2015; Wang et al., 2007; Yu et al., 2013; Zheng et al., 2018; Zinzi et al., 2020). Urban areas usually experience warmer temperatures compared with rural areas (Oke and Maxwell, 1975; Rizwan et al., 2008; Roth, 2007; Stewart and Oke, 2012), and this strong UHI effect can be observed in Beijing all year around (Ren et al., 2007; Wang et al., 2017; Yang et al., 2013). As a result, urban residents are more vulnerable to health risks posed by heat stress. On the other hand, the increased $O_3$ concentration induced by urbanization was found to translate to a 39.6% increase in premature mortality (Yim et al., 2019).

Importantly, whilst urban greening might alleviate the UHI effect (Doick et al., 2014; Zhou et al., 2019), the additional biogenic volatile organic compounds (VOCs) emitted from vegetation under high-temperature stress can favour $O_3$ production (Ma et al., 2019; Wang et al., 2021a; Werner et al., 2020). For example, Ma et al. (2019) found that the landscape in Beijing yields an extra 4.47 ppbv of MDA8 $O_3$ (the maximum daily 8-h average concentration of $O_3$) due to the increase in urban isoprene emissions associated with HWs, and the isoprene emitted from forests in rural areas should also not be underestimated. But

what roles do synoptic weather patterns and urbanization play in the formation of complex HW–$O_3$ compound events in Beijing



with its high population density? What are the public health effects caused by these compound events? These important scientific issues warrant further investigation.

Therefore, the present study on extreme HW–O$_3$ compound events that occurred in Beijing during summertime of 2014–2019 was carried out to comprehensively investigate the roles of synoptic weather patterns and urbanization in these joint events based on surface observation and reanalysis data. Then, the contributions of weather types and urbanization to the excess mortality brought about by combined heat and O$_3$ stress were quantified according to the established health assessment model of Liu et al. (2021) and Yin et al. (2017). The findings are expected to provide a scientific reference for the monitoring and forecasting of summertime HW–O$_3$ compound events and their health risks in Beijing from the perspective of synoptic patterns and urbanization.

## 2 Data and Methods

### 2.1 Data

Ground-level O$_3$ observation data during summertime (June–August) of 2014–2019 were retrieved from Beijing Municipal Ecological and Environmental Monitoring Center, ultimately comprising 31 air quality stations (11 for urban stations, 11 for rural stations, three for traffic stations, and six for other stations) with a missing-values rate for the O$_3$ concentration of less than 10%.

In order to better assess the relationship between O$_3$ pollution and the meteorological variables, we selected 29 automatic weather stations (AWSs) closest to the environmental monitoring stations from the high-density AWS network (Figure 1; Table 1). Hourly 2-m air temperature, relative humidity (RH), the daily maximum temperature (T$_{max}$), and 10-m wind speed (WS) were obtained from the National Meteorological Information Center of the China Meteorological Administration, and then heat index (HI) was retrieved as shown in Rothfusz (1990) as Eq. (1):

$$HI = -42.379 + 2.04901523 \times T + 10.14333127 \times RH - 0.22475541 \times T \times RH - 0.00683783 \times T^2 - 0.05481717 \times RH^2 + 0.00122874 \times T^2 \times RH + 0.00085282 \times T \times RH^2 - 0.00000199 \times T^2 \times RH^2 , \quad (1)$$

Where $T$ indicates the temperature (unit: °F), and $RH$ (unit: %) indicates relative humidity.

In addition, we also used the hourly geopotential height (GH), boundary layer height (BLH), wind vector, vertical velocity and temperature fields to further analyze the weather type and local boundary layer characteristics under the joint occurrence of HW and O$_3$ pollution (Fifth major global reanalysis produced by the European Centre for Medium-Range Weather Forecasts, with spatiotemporal resolution of 0.25°).



## 2.2 Methods

### 2.2.1 Compound HW and O₃ pollution events

An HW event is usually characterized by the daily maximum temperature reaching or exceeding a certain threshold (it can be a relative value or an absolute threshold) for several consecutive days (Ngarambe et al., 2020). In this paper, we selected 33°C (which corresponds to the 90$^{th}$ percentile of $T_{max}$ during 2014–2019 in Beijing) as threshold for $T_{max}$ lasting for 3 days or more to determine an HW event; otherwise, it was a non-heat wave (NHW) event. Moreover, the occurrence of precipitation during the day inhibits the photochemical reaction of O₃ production (Yu et al., 2020; Zhang et al., 2015; Zhao and Wang, 2017), here
a daytime precipitation event (accumulated precipitation ≥ 2 mm during 0700–1900 LST) was excluded to avoid the impact of precipitation on compound HW and O₃ pollution events. O₃ pollution was identified as when the MDA8 O₃ concentration exceeded 160 µg m⁻³, which is in accordance with the Ambient Air Quality Standards issued by the Ministry of Ecology and Environment of the People's Republic of China. Based on the above criteria, 84 days of co-occurring HW and O₃ pollution events during 2014–2019 were finally obtained.

### 2.2.2 Weather type classification

According to previous studies (Han et al., 2020; Miao et al., 2019; Ning et al., 2019; Yang et al., 2018, 2021; Zhang and Villarini, 2019), T-mode principal component analysis (T-PCA) was applied to classify the 850-hPa GH field of the joint occurrence of HW and O₃ pollution events [more specific details on T-PCA were introduced in our previous study (Zong, et al., 2021a)]. As for the categorical data, we mainly focused on the domain (110°–125°E, 32°–47°N), including Beijing,
associated with these 84 days of compound events during summertime (June–August) 2014–2019.

### 2.2.2 Excess mortality

In epidemiology, the relative risk (RR) is usually used to evaluate the intensity of the association between exposure and disease, which refers to the ratio of the incidence of the exposed group to the incidence of the non-exposed group (Chen et al., 2018; Pope et al., 2016). The RR is calculated by Eq. (2):

$$RR_i = exp^{\beta_i * \Delta X_i}, \tag{2}$$

where $i$ indicates the risk factor (high temperature or O₃ concentration), $\beta_i$ is the exposure response function between the risk factor $i$ and total mortality through nonlinear regression (Cao et al., 2021; Du et al., 2020; Gu et al., 2018), $\Delta X_i$ is the difference between the risk factor $i$ and its reference health threshold. The excess risk (ER) is calculated by Eq. (3):

$$ER_i = (RR_i - 1) \times 100\%, \tag{3}$$

Here, we refer to the exposure response function for the high temperature as suggested by Liu et al. (2021), and O₃ concentration as suggested by Yin et al. (2017) in northern China. In detail, Liu et al. (2021) investigated the mortality caused by high temperature in 84 cities in China from 2013 to 2016, and found that for every 1°C increase in the daily $T_{max}$ above 31.5°C, the largest RR of mortality caused by high temperature in northern China was 1.002 [95% confidence interval (CI):





1.001, 1.004]. For $O_3$ exposure, a 10-µg m$^{-3}$ increase in MDA8 $O_3$ was related to an increase in the total daily mortality of 0.39% (95% CI: 0.04%, 0.75%) in northern China during the warm season (Yin et al., 2017). Since the two models have removed the mutual influence, the final joint ER is the sum of the ERs of both high temperature and $O_3$.

### 2.2.3 Urbanization and Synoptic contribution rates

To estimate the impact of urbanization and weather patterns on compound HW and $O_3$ pollution events, we further determined their contribution rates to the excess mortality of compound events. With reference to Ma & Yuan (2021), the urbanization

effect is calculated by Eq. (4):

$$\Delta ER_{i,urbanization} = ER_{i,urban} - ER_{i,rural}, \tag{4}$$

and contribution rate is calculated by Eq. (5):

$$CR_{i,urbanization} = \frac{\Delta ER_i}{ER_{i,urban}} \times 100\%, \tag{5}$$

Where $i$ indicates risk factor (high temperature or ozone pollution), ER is excess mortality, and CR is contribution rate.

Similarly, we also defined synoptic effects as Eq. (6):

$$\Delta ER_{i,synoptic} = ER_{i,synoptic} - ER_{i,average}, \tag{6}$$

and the contribution rate as Eq. (7):

$$CR_{i,synoptic} = \frac{\Delta ER_i}{ER_{i,synoptic}} \times 100\%, \tag{7}$$

Where $i$, ER, and CR are same as Eq. (6).

## 3 Results

### 3.1 Compound HW–O₃ pollution events and associated public health in Beijing

Figure 2 shows the time series of the HW days, NHW days, $O_3$ pollution, and precipitation days, showing that HW periods (approximately 79.2% of HW days, Figure 3b) in Beijing, were always paired above-threshold $O_3$ pollution levels. Daytime precipitation obviously inhibits the photochemical reaction with consequent fewer $O_3$ pollution episodes occurred during

daytime precipitation (the average 24-h $O_3$ concentration was reduced by 17.14 µg m$^{-3}$ on precipitation days compared with NHW days). Note that there was an increase in the maximum duration of HW events and the number of HW–$O_3$ paired days during summertime of 2014–2019 (Figure 3, by 2019, the most durable HW event lasted for 10 days), resulting in people suffering from more enduring dual heat and $O_3$ stress. As shown in Figure 4, MDA8 $O_3$ aggravated significantly on HW days, exceeding 160 µg m$^{-3}$ across all stations, with an average of 189.35 µg m$^{-3}$. Nevertheless, also during NHW days, some MDA8

$O_3$ stations (mainly urban or vegetation covered stations) exceeded the pollution threshold. Both surface $O_3$ concentration and MDA8 $O_3$ concentration in Beijing showed significant differences ($P < 0.01$) among three conditions (Table 2). We further investigated the diurnal variation for surface air temperature (T), RH, HI, BLH and WS under HW, NHW and precipitation



periods (Figure 5), and these five variables also showed significant differences in the three periods. For HW days, HI was aggravated more by increased air temperature, and although the RH was relative lower, people still suffered from higher apparent temperature than actual air temperature. During HW conditions, solar radiation reaching the ground heats the atmosphere increasing the near-surface temperature. Warmer air convection promotes atmospheric instability, with increased WS and higher BLH. It is clear that the meteorological variables during daytime HW periods were significantly different than under other conditions; and similarly, hourly $O_3$ concentrations under these conditions reached the peaks (Figure 5f). Therefore, it can be concluded that the difference in $O_3$ concentration was mainly due to meteorological conditions paired with photochemical reactions in the boundary layer without considering $O_3$ emission precursors.

Moreover, the high temperatures on HW days not only brought a higher public risk related to high-temperature exposure, but also an increase in mortality related to $O_3$ exposure. Regardless of the type of stations, the $O_3$ and high-temperature stressful conditions suffered by the population during HW days has greatly increased. Specifically, for all stations, HWs have increased the ER caused by high temperatures and $O_3$ by 3.867% (90% CI: 3.863%, 3.875%) and 7.9% (90%CI: 0.78%, 15.78%), respectively (Table 3). Interestingly, urbanization seems to have increased the ER induced by high temperatures over urban stations by 39.88%, but caused a 2.44% reduction in the ER associated with $O_3$. On the one hand, high temperatures and strong solar radiation during HW periods accelerate the rate of the photochemical reaction that produces $O_3$ (Pu et al., 2017; Sun et al., 2017), which is conducive to the production and accumulation of $O_3$. On the other hand, they increase the VOC emission of plants and contributes to the production of $O_3$ (Ma et al., 2019; Trainer et al., 1987; Wang et al., 2021a). The majority of the isoprene emissions from forests in Beijing are mainly located in rural areas (Ma et al., 2019). Therefore, the $O_3$ concentrations over rural stations were higher than those over urban stations. In addition, the risk of $O_3$ exposure in the traffic stations was significantly lower than that of the urban and rural stations, and the risk of high-temperature exposure was slightly lower than that of urban stations. This has also led to the overall paired high-temperature and $O_3$ risk over traffic stations being lower than that over rural stations.

## 3.2 Role of synoptic weather pattern and urbanization

To further clarify the mechanism underlying the joint occurrence of HW–$O_3$ events in Beijing, three unfavourable synoptic weather patterns were identified as follows: (1) Type 1, characterized by the western Pacific subtropical high (WPSH) being located in the southeast of Beijing with prevailing southwesterly winds; (2) Type 2, controlled by a high-pressure system accompanied by weak southerly winds; and (3) Type 3, a low-vortex located over northeast Beijing with prevailing northwesterly winds (Figures 6a–6c). Additionally, vertical cross-sections of the potential temperature and wind vectors at 1400 LST under the three patterns are shown in Figures 6e–6f. Under Type 1, low boundary layer paired with weak vertical motion favours pollutants' accumulation. Besides, the prevailing southwesterly wind may blow pollutants from the upwind direction to Beijing, and the northern mountains block the pollutants from continuing to be transported in the downward wind direction, causing the pollutants to gather in Beijing. For Type 2, a lower BLH and vertical convection together regulate the transportation and accumulation of $O_3$ in the boundary layer. Under Type 3, there is a valley–plain wind circulation in the



boundary layer, and the strong downdraft over urban areas and the higher boundary layer cause the lowest MDA8 $O_3$ concentrations among the three weather types. Overall, Type 1 tends to be associated with the highest excess mortality caused by $O_3$, and Type 3 is related with the highest excess mortality caused by HWs. For excess mortality induced by the HW–$O_3$ compound events, Type 1 (12.59%) > Type 3 (12.05%) > Type 2 (11.66%). Although there is little difference in the HW–$O_3$

compound ER under the three weather types, the mechanisms of the three types are quite different. Under the modulation of weather circulation and boundary layer meteorological elements, Type 1, Type 2 and Type 3 were associated with high $O_3$ and intermediate $T_{max}$ exposure, intermediate $O_3$ and low $T_{max}$ exposure, and low $O_3$ and high $T_{max}$ exposure (Figure 7), respectively. Therefore, the synoptic weather pattern plays an important role in regulating the formation mechanism of HW–$O_3$ compound events, which also further leads to it having a significant impact on morbidities and deaths caused by HW–$O_3$ compound events.

Table 4 also shows that there was an opposite urbanization regulation effect between high temperature and $O_3$ and their health risks under Type 1. In particular, HWs extended the urban–rural air temperature difference (the UHI effect) in Beijing (Table 2), as also found in our previous study (Zong et al., 2021b). It was mentioned earlier that the $O_3$ pollution in rural areas is higher than that in urban areas during HW periods. For Type 1, the strong southerly airflow caused the horizontal transportation of $O_3$ and its precursors from the southwest (urban) to the northeast (rural). Paired with the large number of forests in rural

areas during the HW that emitted additional VOCs (Ma et al., 2019), this further increased the difference in the $O_3$ concentration between urban and rural areas. Urbanization has a positive regulatory effect on the risk of heatwaves, but a negative one with respect to the risk of $O_3$. As for Type 2 and Type3, there was a little difference in $O_3$ exposure risk between urban and rural areas, but urban residents were more likely to be exposed to higher-temperature environment under Type 3 compared to Type 2. Specifically, the contribution rates of urbanization to the excessive mortality caused by high temperatures

and $O_3$ exposure were 27.72% and −2.58%, respectively, while the contribution rates of the synoptic pattern were 85.54% and 24.00%. In summary, urbanization and the synoptic pattern contributed 8.08% and 45% to the total HW–$O_3$ excess mortality (Table 5).

## 4 Discussions

In addition to heatstroke, heat exhaustion, heat fainting and heat cramps and other diseases, high temperature during HW days

can also lead to increased mortality of residents. Several studies have proposed that the mortality because of respiratory diseases, cardiovascular diseases and cardiopulmonary diseases induced by high temperature and $O_3$ exposure is particularly relevant (Chen et al., 2018; Du et al., 2020; Hu et al., 2019). Therefore, patients with pre-existing conditions as respiratory and cardiovascular diseases, should pay more attention and limit outdoor activity under heatwaves and $O_3$ polluted days. Furthermore, demographic, and socio-economic factors related to the level of urbanization, including age structure, education

and healthcare services, occupational types, and air-conditioning use, also greatly affect the exposure response function of high temperature and $O_3$. For instance, females, elderly and people with a lower degree of instruction have suffered significantly higher health risks from overexposure of high temperature and $O_3$ than the average population (Huang et al., 2015;





Yin et al., 2017; Zhang et al., 2017). However, this study mainly considers mortality by all-causes for all the population caused by high temperature and O$_3$ exposure. Health risks for high-risk groups such as the elderly, children, and patients with
cardiovascular and respiratory diseases should be higher than our results. Consequently, especially during synoptic weather patterns that can cause paired HW and O$_3$ pollution events, the responsible departments should strengthen the risk management of extreme compound events such as HW and O$_3$ pollution, establish an early warning system, configure emergency plans, strengthen the health precautions of respiratory and cardiovascular diseases, as well as the elderly and other vulnerable groups, and protect public health.

To date, there is no exact consensus on urbanization effects on risk of paired high temperature and O$_3$ exposure. Previously, a common perception was that urban residents were more prone to risks of heat effect in the context of global warming and UHI effect (Clarke, 1972; Goggins et al., 2012; Heaviside et al., 2017). Indeed, air temperature is one of the main reasons dominate the change in excess mortality caused by compound HW and O$_3$ events, In terms of the urban areas, the higher density of buildings, roads and population, greater heat capacity and anthropogenic heat, temperature of urban areas is significantly higher
than that of rural areas (Roth, 2007; Stewart & Oke, 2012). The heatwaves increase the urban and rural areas temperature difference, as well as the maximum temperature difference, so urban residents may expose to a higher temperature environment. However, the urban-rural difference in O$_3$ concentration modulated by HW days is inconsistent with that in temperature. Rural forests emit additional VOCs that generate O$_3$ during hot days (Ma et al., 2019; Wang et al., 2021a; Werner et al., 2020), resulting in O$_3$ pollution slightly lower in urban than rural areas(Gao et al., 2020a). Based on the regional exposure response
function model, urban areas suffer from higher mortality related to high temperature, while rural areas experience higher public mortality associated with O$_3$ pollution. It should be highlighted that also urbanization brought also some positive aspects. For example, a better economic level and medical conditions, can help to prevent more deaths to a certain extent; high air-conditioning utilization rate can also effectively reduce heat exposure; and the reduction in the proportion of highly exposed people engaged in agriculture, forestry and animal husbandry in urban areas also greatly reduces the risk of outdoor high-
temperature and O$_3$ overexposure. As a result, rural residents are more vulnerable to face the dual high temperature and O$_3$ stress, and their exposure response function coefficients may also be higher than that of urban residents (Hu et al., 2019; Kovach et al., 2015; Li et al., 2017; Williams et al., 2013; Xing et al., 2020; Zhang et al., 2017). This also means that under co-occurring heatwaves and ozone-polluted weather patterns, vulnerable groups in the suburbs should be warned on the risks of outdoor activity and limiting their exposure to the pollutants.

Regarding with the paired HW–O$_3$ events, though we moved a step forward in exploring role of synoptic weather pattern and urbanization, there are still some limitations in our study. As mentioned earlier, in a specific area, the health risks faced by residents adjusted by different levels of urbanization may be quite different. Moreover, the high temperature and O$_3$ compound health risk model for special populations (e.g., patients with cardiovascular and respiratory diseases, the elderly, children, etc.) can also be further established and analyzed. Therefore, in the future research on compound climate and pollution health
impacts it is necessary to consider a more refined discussion in a city based on the level of urbanization and among different population groups.



## 5 Conclusions

In this study, the complex mechanism of co-occurring HW–$O_3$ events in the boundary layer in Beijing was systematically investigated by combining meteorological observations, environmental monitoring observations, and reanalysis data, and the

regulatory role on health risks induced by such compound events was explained from the perspective of the synoptic pattern and urbanization.

The Beijing area not only experienced a stronger UHI effect during the summertime high-temperature HWs, but was also often accompanied by more serious $O_3$ pollution. In the period under study, the max temperature $T_{max}$ and MDA8 $O_3$ concentrations during HW days were ~4.21℃ and ~37.98 μg m$^{-3}$ higher than those on NHW days, respectively, excluding rainy daytime

days. When people are exposed to the dual stress of high temperatures and $O_3$ pollution on HW days, the increase in $T_{max}$ and MDA8 $O_3$ concentrations is associated with an 11.78% (95% CI: 4.66%, 19.66%) higher excess mortality from all non-accidental causes. Three unfavourable synoptic weather patterns that dominate such compound events and were identified as: (1) Type 1, a high-pressure system located in the southeast of Beijing and accompanied by southwesterly winds, under which the weak downdraft and relative stable boundary layer weaken the vertical mixing of $O_3$ and induce heavy $O_3$ pollution,

consequently meaning that people consistently experience high health risks; (2) Type 2, in which Beijing is under the influence of a high-pressure system accompanied by weak southerly winds and sinking airflow in the boundary layer that favours $O_3$ transport together with its precursors. This translates into a lower excess mortality under Type 2 with respect to Type 1; and (3) Type 3, a low-pressure system located in the northeast of Beijing accompanied by northwesterly winds. Under this type, people endure stronger heat stress owing to higher temperatures and lower RH, but the higher BLH and large atmospheric

environment capacity alleviate $O_3$ exposure to a certain extent, which results in a decrease in $O_3$ concentration and ER compared with the other two patterns. Overall, the unfavourable weather types contributed ~45.76% to the excess mortality attributed to the HW–$O_3$ compound events.

In addition, urbanization has also exacerbated the combined health risks of high temperature and $O_3$ pollution, which contributed ~8.08%. During the co-occurring HW–$O_3$ days, urbanization greatly affected the increase in high temperatures

and related excess mortality risks in urban areas, which were significantly higher than those in rural areas. On the contrary, $O_3$ pollution and its health risks in urban areas were lower than those in rural areas, and the contribution of urbanization effects was negative. This is because those urban pollutants ($O_3$ and its precursors) were diffused and transported to the rural areas via local urban island circulation, which greatly increased the $O_3$ pollution and resultant excess mortality risk in rural areas, and reduced the $O_3$ concentration and related health risks in urban areas.

In summary, our findings help to better understand the formation mechanism of HW–$O_3$ compound events in Beijing, with robust supporting evidence from the perspective of synoptic patterns and urbanization. Our results also suggest that forecasting of identified synoptic patterns could help to avoid exposure of compound HW-$O_3$ events. However, the urbanization effect has an opposite regulatory effect on HWs and $O_3$, meaning that high temperatures and $O_3$ exposure is deserving of the establishment of a more refined health model that takes into account the differences between urban and rural areas.





## Data availability

The datasets that are analyzed and used to support the findings of this study are available in the public domains: The ground-level O3 observation data can be obtained from Beijing Municipal Ecological and Environmental Monitoring Center (http://www.bjmemc.com.cn/, last access on November 20, 2021). The hourly meteorological data can be obtained from the National Meteorological Information Center of the China Meteorological Administration (http://data.cma.cn/, last access on November 20, 2021). The ERA5 reanalysis data set is available at the European Centre for Medium-Range Weather Forecasts (https://cds.climate.copernicus.eu/cdsapp#!/home, last access on November 20, 2021).

## Competing interests

The authors declare that they have no conflict of interests.

## Author contributions

L. Zong: Methodology, Data Curation, Formal Analysis, Writing- Original draft preparation, Results Discussion, Writing-Reviewing and Editing; Y. Yang, H. Xia: Conceptualization, Methodology, Formal Analysis, Results Discussion, Writing-Reviewing and Editing; Z. Sun, Z. Zheng, X. Li, G. Ning, Y. Li, Z. Gao, S. Lolli: Results Discussion, Comments, Writing-Reviewing and Editing.

## Acknowledgments

This research was supported by the National Natural Science Foundation of China (42175098).

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





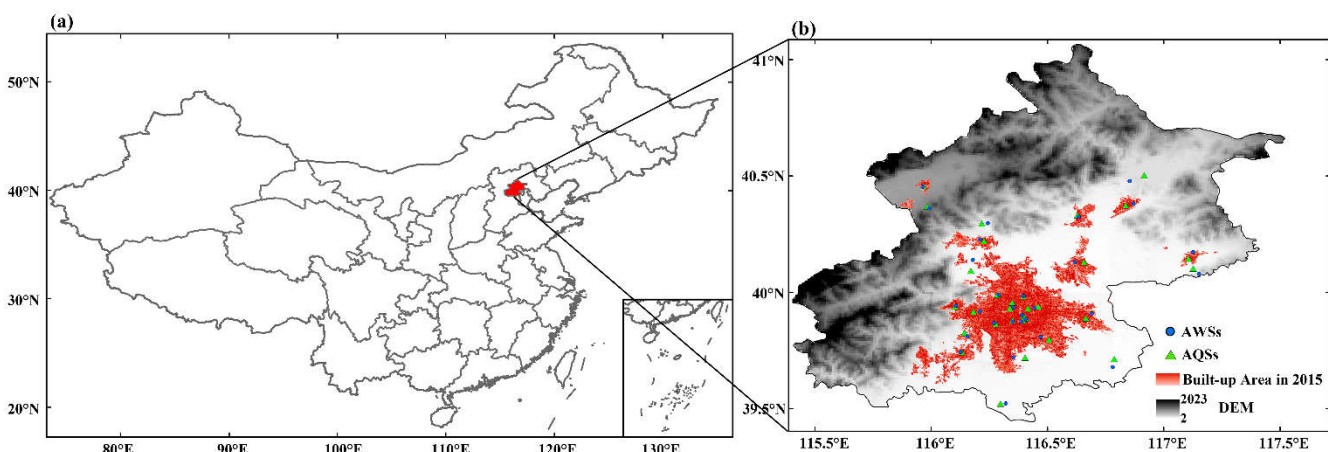


**Figure 1: (a) Geography of Beijing. (b) Distribution of AWSs and air quality stations (AQSs) in Beijing (superimposed on the built-up area data for 2015 from digital elevation model data).**





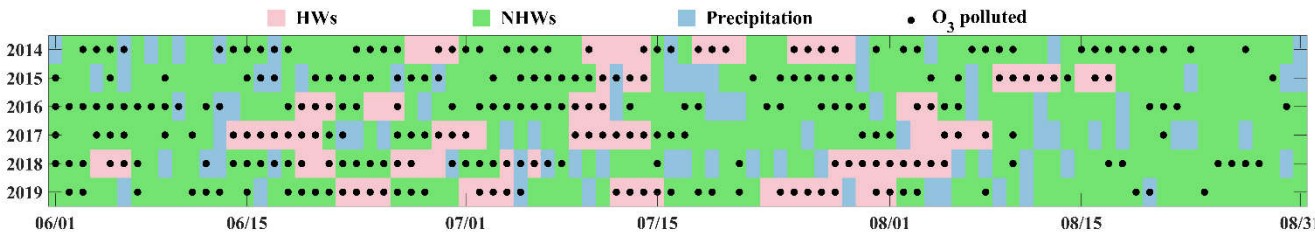

**Figure 2: Time series of weather types, in which the black dots indicate O₃ pollution that occurred on that day.**





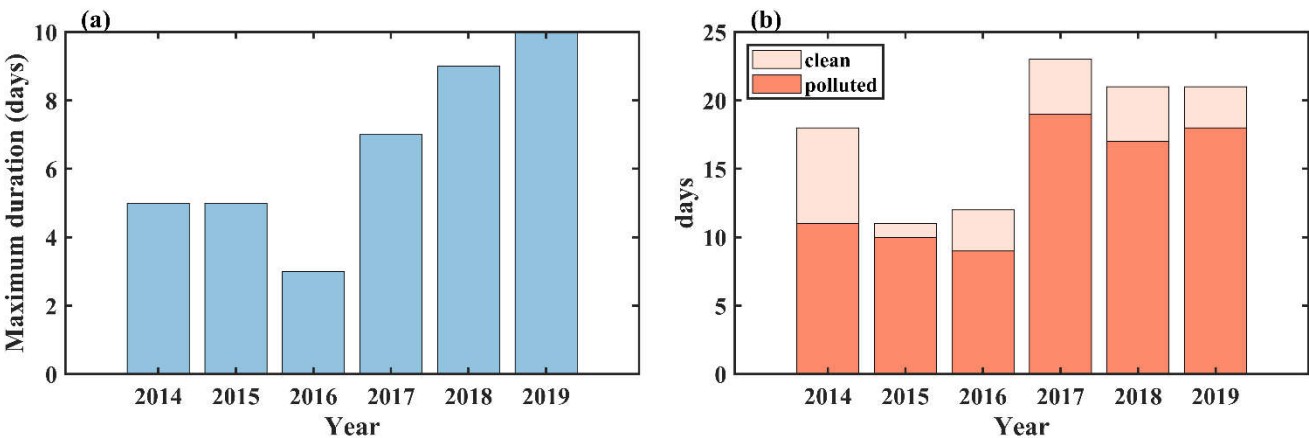

**Figure 3: (a) Maximum number of days of HW events each year. (b) Proportion of O₃ pollution during HW events each year.**




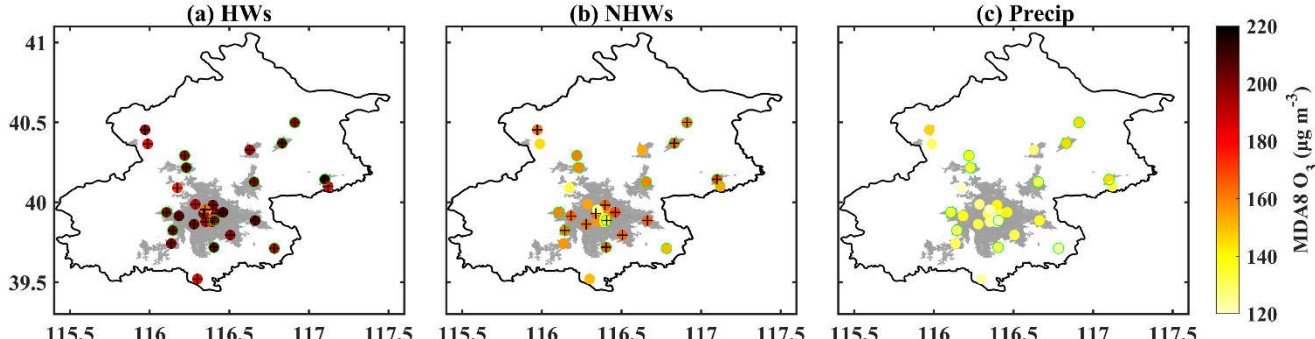

**Figure 4: Distribution of MDA8 O₃ under (a) HWs, (b) NHWs, and (c) precipitation periods (superimposed on built-up area data for 2015, with green dots indicating vegetation covered stations). The diurnal variation of (d) air temperature, (e) RH, (f) HI, (g) BLH, (h) WS and (i) O₃, under HWs, NHWs and precipitation periods (shading indicates standard deviation).**




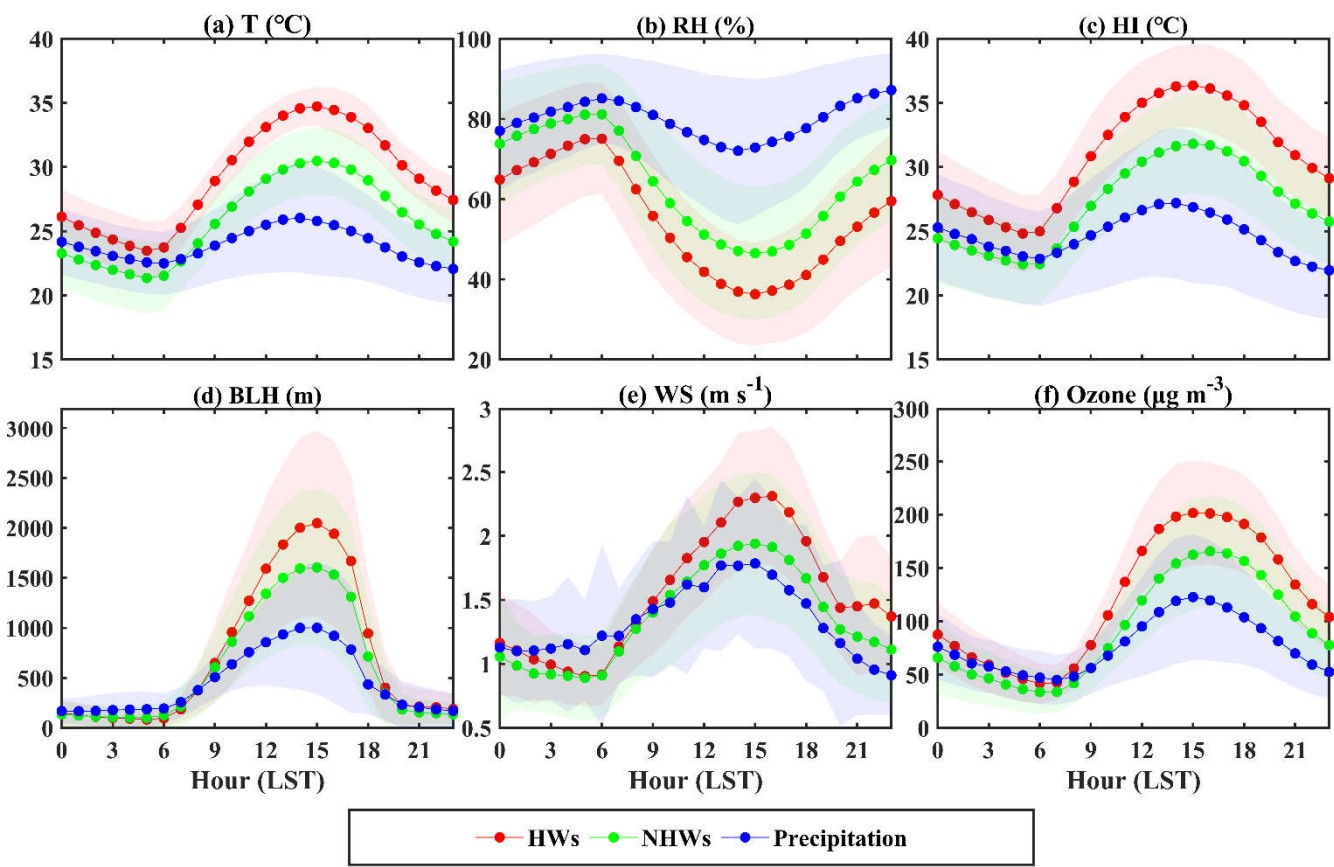

**Figure 5: The diurnal variation of (a) air temperature, (b) RH, (c) HI, (d) BLH, (e) WS and (f) O₃, under HWs, NHWs and precipitation periods (shading indicates standard deviation).**



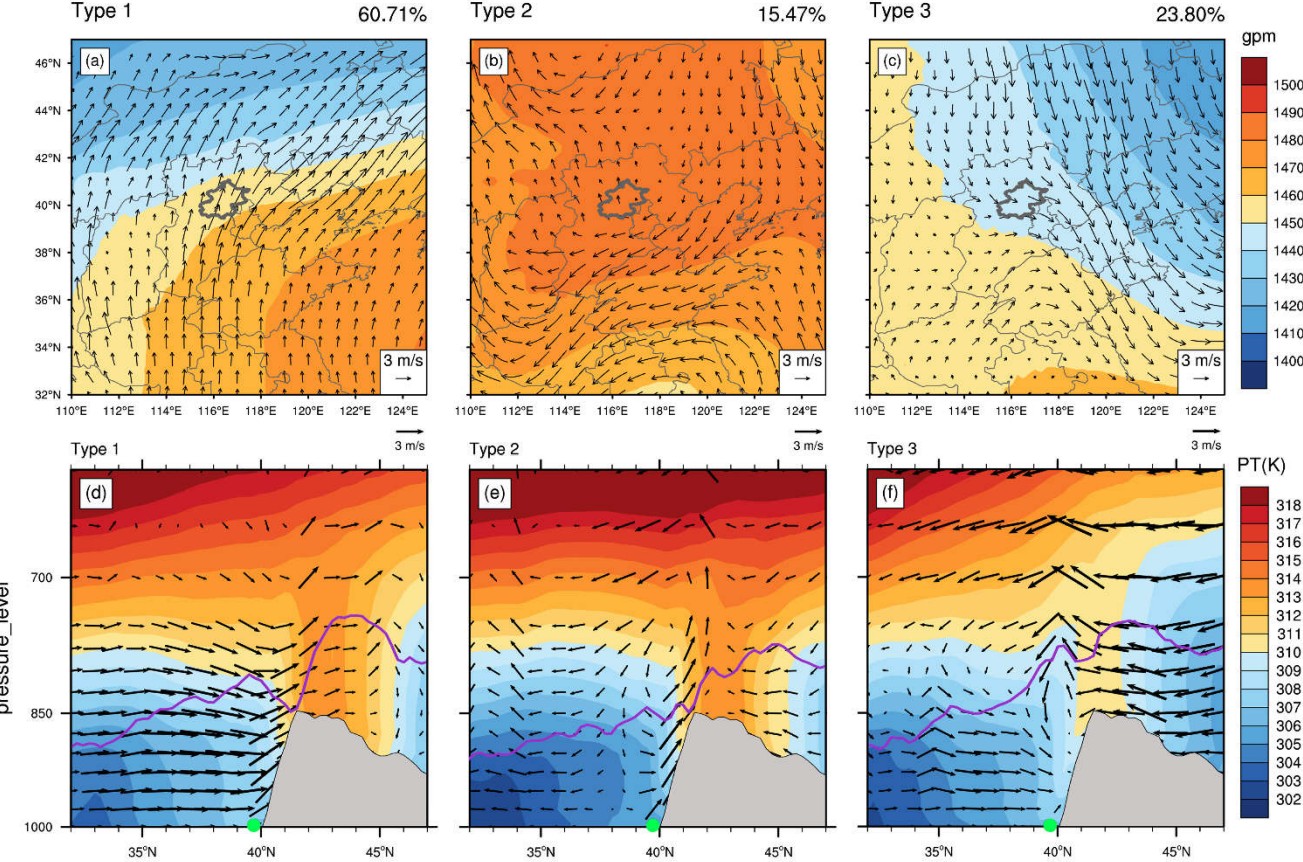

**Figure 6: (a–c) 850-hPa GH (contours) and υv-wind (vectors) patterns related to HWs and O₃ pollution compound events based on objective classification (grey outline represents Beijing, and the number in the upper-right corner of each panel indicates the frequency of occurrence of each pattern). (d–f) Vertical cross-sections of the potential temperature (contours) and wind vectors (synthesized by v and scaled ω, ω scaled by 100) averaged between 116.0°E and 117.0°E associated with the synoptic patterns (purple solid lines mark the BLH, black contours mark the topography, and the green dot marks the location of Beijing).**





**Figure 7: Schematic illustration of the mechanism of HW and O₃ compound pollution events under different synoptic weather patterns (height of the icon indicates the size of each variable).**





**Table 1: The location information and station type of AQSs, the corresponding AWSs are the closest matching weather station among 295 AWSs.**

| AQI Station | Lon (°) | Lat (°) | Type | AWS | Lon (°) | Lat (°) |
|---|---|---|---|---|---|---|
| DS | 116.42 | 39.93 | Urban | A1003 | 116.44 | 39.93 |
| TT | 116.41 | 39.89 | Urban* | A1016 | 116.41 | 39.88 |
| GY | 116.34 | 39.93 | Urban | A1006 | 116.35 | 39.93 |
| WSXG | 116.35 | 39.88 | Urban | A1015 | 116.35 | 39.87 |
| ATZX | 116.40 | 39.98 | Urban | A1007 | 116.40 | 39.98 |
| NZG | 116.46 | 39.94 | Urban | A1003 | 116.44 | 39.93 |
| WL | 116.29 | 39.99 | Urban | 54399 | 116.29 | 39.99 |
| BBXQ | 116.17 | 40.09 | Urban | A1033 | 116.18 | 40.14 |
| FTHY | 116.28 | 39.86 | Urban | A1053 | 116.27 | 39.87 |
| YG | 116.15 | 39.82 | Urban* | A1037 | 116.16 | 39.81 |
| GC | 116.18 | 39.91 | Urban | A1019 | 116.21 | 39.92 |
| FS | 116.14 | 39.74 | Rural | A1314 | 116.13 | 39.74 |
| DX | 116.40 | 39.72 | Rural* | 54594 | 116.35 | 39.72 |
| YZ | 116.51 | 39.80 | Rural | 54511 | 116.47 | 39.81 |
| TZ | 116.66 | 39.89 | Rural | A1213 | 116.69 | 39.91 |
| SY | 116.66 | 40.13 | Rural* | 54398 | 116.62 | 40.13 |
| CP | 116.23 | 40.22 | Rural* | 54499 | 116.21 | 40.22 |
| MTG | 116.11 | 39.94 | Rural* | A1354 | 116.11 | 39.94 |
| PG | 117.10 | 40.14 | Rural* | 54424 | 117.12 | 40.17 |
| HR | 116.63 | 40.33 | Rural | A1621 | 116.63 | 40.32 |
| MY | 116.83 | 40.37 | Rural* | 54416 | 116.86 | 40.38 |
| YQ | 115.97 | 40.45 | Rural | 54406 | 115.97 | 40.45 |
| QM | 116.40 | 39.90 | Traffic | A1001 | 116.39 | 39.90 |
| YDM | 116.39 | 39.88 | Traffic | A1020 | 116.39 | 39.87 |
| XZMB | 116.35 | 39.95 | Traffic | A1006 | 116.35 | 39.93 |
| DL | 116.22 | 40.29 | Other* | A1407 | 116.25 | 40.29 |
| BDL | 115.99 | 40.37 | Other | A1468 | 116.00 | 40.36 |
| MYSK | 116.91 | 40.50 | Other* | A1655 | 116.85 | 40.47 |
| DGC | 117.12 | 40.10 | Other* | A1514 | 117.14 | 40.08 |
| YLD | 116.78 | 39.71 | Other | A1201 | 116.78 | 39.68 |
| YF | 116.30 | 39.52 | Other | A1252 | 116.32 | 39.52 |

**Note: The asterisk in the type column indicates that the underlying surface of the observing station is covered by**
**vegetation.**



**Table 2: Analysis of variance of each variable under different weather conditions (HWs, NHWs and precipitation).**

| | Source | SS | df | MS | F | P |
|---|---|---|---|---|---|---|
| **Tmax** | Source | SS | df | MS | F | P |
| | Group | 2609.52 | 2 | 1304.76 | 215.27 | 1.09E-69 |
| | Error | 3315.38 | 547 | 6.06 | | |
| | Total | 5924.89 | 549 | | | |
| **MDA8O$_3$** | Source | SS | df | MS | F | P |
| | Group | 204454.6 | 2 | 102227.3 | 56.32 | 6.03E-23 |
| | Error | 981941.2 | 541 | 1815 | | |
| | Total | 1186396 | 543 | | | |
| **ER$_{HW}$** | Source | SS | df | MS | F | P |
| | Group | 2614.65 | 2 | 1307.33 | 221.48 | 3.46E-71 |
| | Error | 3228.8 | 547 | 5.9 | | |
| | Total | 5843.45 | 549 | | | |
| **ER$_{ozone}$** | Source | SS | df | MS | F | P |
| | Group | 353.62 | 2 | 176.808 | 56.4 | 5.63E-23 |
| | Error | 1695.85 | 541 | 3.135 | | |
| | Total | 2049.47 | 543 | | | |
| **ER$_{total}$** | Source | SS | df | MS | F | P |
| | Group | 4867.9 | 2 | 2433.94 | 207.28 | 1.69E-67 |
| | Error | 6329.2 | 539 | 11.74 | | |
| | Total | 11197.1 | 541 | | | |





**Table 3: RH, temperature, HI, MDA8 O₃, O₃ concentration, and ER of HW and O₃ pollution compound events for mortalities in different station types associated with different weather conditions.**

| Station type | Period | RH (%) | $T_{mean}$ (°C) | $T_{min}$ (°C) | $T_{max}$ (°C) | $HI_{mean}$ (°C) | $HI_{min}$ (°C) | $HI_{max}$ (°C) | MDA8 O₃ (µg m⁻³) | O₃ mean (µg m⁻³) | $ER_{HW}$ (%) | $ER_{Ozone}$ (%) | $ER_{total}$ (%) |
|---|---|---|---|---|---|---|---|---|---|---|---|---|---|
| Urban | HWs | 53.8 | 30.1 | 24.0 | 36.1 | 32.0 | 25.3 | 38.3 | 197.1 | 119.8 | 4.76(4.76,4.77) | 8.01(0.79,16.00) | 12.78(5.56,20.78) |
| | NHWs | 62.4 | 26.6 | 21.6 | 31.8 | 27.8 | 22.0 | 33.3 | 158.5 | 91.5 | 0.3179(0.3175,0.3186) | 6.40(0.64,12.70) | 6.69(0.94,12.99) |
| | Precip | 80.1 | 24.4 | 21.4 | 28.5 | 25.1 | 19.8 | 31.0 | 130.6 | 73.5 | -2.868 (-2.866,-2.87) | 5.24(0.52,10.36) | 2.38(-2.34,7.49) |
| Rural | HWs | 56.7 | 29.0 | 23.0 | 34.8 | 30.8 | 24.0 | 36.8 | 201.8 | 126.5 | 3.403(3.400,3.410) | 8.21(0.81,16.42) | 11.61(4.22,19.83) |
| | NHWs | 64.5 | 25.7 | 20.7 | 30.7 | 26.9 | 21.3 | 32.0 | 161.0 | 96.6 | -0.798(-0.797,-0.799) | 6.50(0.65,12.92) | 5.67(-0.17,12.08) |
| | Precip | 79.1 | 23.9 | 20.8 | 27.8 | 24.7 | 19.7 | 30.0 | 135.7 | 79.4 | -3.553(-3.550,-3.561) | 5.45(0.54,10.79) | 1.88(-3.02,7.21) |
| Traffic | HWs | 50.3 | 30.4 | 25.2 | 35.8 | 32.1 | 26.7 | 37.6 | 169.7 | 118.9 | 4.412(4.408,4.421) | 6.86(0.68,13.63) | 11.28(5.09,18.06) |
| | NHWs | 59.0 | 26.9 | 22.4 | 31.6 | 28.2 | 23.3 | 32.9 | 132.8 | 93.8 | 0.1092(0.1090,0.1095) | 5.33(0.53,10.53) | 5.42(0.63,10.61) |
| | Precip | 77.3 | 24.5 | 21.6 | 28.5 | 25.3 | 20.3 | 30.6 | 106.2 | 78.7 | -2.947(-2.944,-2.952) | 4.24(0.43,8.35) | 1.28(-2.54,5.38) |
| All | HWs | 55.5 | 29.4 | 23.5 | 35.2 | 31.2 | 24.6 | 37.4 | 189.4 | 117.4 | 3.867(3.863,3.875) | 7.90(0.78,15.78) | 11.78(4.66,19.66) |
| | NHWs | 63.8 | 25.9 | 21.1 | 31.0 | 27.2 | 21.6 | 32.6 | 151.4 | 89.6 | -0.4179(-0.4175,-0.4187) | 6.29(0.63,12.49) | 5.85(0.2,12.03) |
| | Precip | 79.7 | 24.0 | 20.9 | 28.0 | 24.7 | 19.7 | 30.4 | 125.9 | 72.5 | -3.377(-3.374,-3.384) | 5.23(0.52,10.33) | 1.84(-2.86,6.94) |
| Ur–Ru | HWs | -2.9 | 1.0 | 1.1 | 1.3 | 1.2 | 1.3 | 1.5 | -4.7 | -6.7 | 1.36 | -0.20 | 1.17 |
| | NHWs | -2.2 | 0.9 | 0.9 | 1.1 | 0.9 | 0.7 | 1.3 | -2.5 | -5.1 | 1.12 | -0.10 | 1.02 |
| | Precip | 0.9 | 0.5 | 0.5 | 0.7 | 0.4 | 0.1 | 1.0 | -5.1 | -5.9 | 0.08 | -0.21 | 0.50 |

**Note: Ur–Ru: Urban–Rural.**





**Table 4: RH, temperature, HI, MDA8 O$_3$, O$_3$ concentration, and ER of HW and O$_3$ compound pollution events for mortalities at different station types associated with different synoptic patterns.**

| Station type | Period | RH (%) | T$_{mean}$ (°C) | T$_{min}$ (°C) | T$_{max}$ (°C) | HI$_{mean}$ (°C) | HI$_{min}$ (°C) | HI$_{max}$ (°C) | MDA8 O$_3$ (μg m$^{-3}$) | O$_3$ mean (μg m$^{-3}$) | ER$_{HW}$ (%) | ER$_{Ozone}$(%) | ER$_{compound}$(%) |
|---|---|---|---|---|---|---|---|---|---|---|---|---|---|
| **Urban** | Type 1 | 55.9 | 30.3 | 24.6 | 36.0 | 32.8 | 26.0 | 39.2 | 216.3 | 133.1 | 4.65(4.64,4.66) | 8.81(0.87,17.66) | 13.45(5.50,22.30) |
|  | Type 2 | 64.0 | 30.5 | 25.5 | 35.7 | 34.5 | 27.7 | 40.8 | 206.2 | 121.6 | 4.40(4.39,4.41) | 8.38(0.83,16.76) | 12.78(5.22,21.16) |
|  | Type 3 | 49.7 | 29.5 | 22.6 | 36.4 | 29.9 | 23.2 | 36.4 | 201.4 | 122.4 | 5.06(5.05,5.07) | 8.18(0.81,16.34) | 13.24(5.86,21.41) |
| **Rural** | Type 1 | 58.2 | 29.4 | 23.6 | 34.8 | 31.6 | 24.8 | 37.7 | 225.6 | 142.6 | 3.381(3.378,3.388) | 9.21(0.91,18.49) | 12.59(4.28,21.88) |
|  | Type 2 | 66.7 | 29.5 | 24.2 | 34.7 | 33.0 | 25.4 | 39.4 | 202.8 | 123.3 | 3.278(3.275,3.285) | 8.24(0.81,16.46) | 11.53(4.09,19.77) |
|  | Type 3 | 53.0 | 28.4 | 21.5 | 35.0 | 28.9 | 22.0 | 35.0 | 201.3 | 127.4 | 3.538(3.534,3.545) | 8.18(0.81,16.35) | 11.71(4.33,19.89) |
| **Traffic** | Type 1 | 52.2 | 30.8 | 25.8 | 35.7 | 32.9 | 27.5 | 38.4 | 185.9 | 116.3 | 4.341(4.337,4.350) | 7.53(0.75,15.02) | 11.87(5.08,19.35) |
|  | Type 2 | 59.7 | 30.8 | 26.5 | 35.6 | 34.4 | 29.3 | 39.5 | 175.8 | 101.2 | 4.161(4.157,4.170) | 7.11(0.71,14.13) | 11.24(4.83,18.27) |
|  | Type 3 | 46.6 | 29.7 | 23.5 | 35.8 | 30.1 | 24.6 | 35.7 | 174.8 | 108.6 | 4.462(4.457,4.471) | 7.06(0.70,14.03) | 11.52(5.16,18.5) |
| **All** | Type 1 | 57.2 | 29.7 | 24.1 | 35.2 | 32.0 | 25.4 | 38.3 | 209.3 | 131.3 | 3.796(3.792,3.804) | 8.80(0.87,17.63) | 12.59(4.66,21.42) |
|  | Type 2 | 65.5 | 29.8 | 24.7 | 35.0 | 33.5 | 26.5 | 39.9 | 195.0 | 114.5 | 3.644(3.640,3.651) | 8.00(0.79,15.97) | 11.66(4.44,19.64) |
|  | Type 3 | 51.6 | 28.7 | 22.0 | 35.4 | 29.2 | 22.5 | 35.5 | 192.5 | 120.0 | 4.063(4.059,4.071) | 7.99(0.79,15.95) | 12.05(4.85,20.02) |
| **Ur–Ru** | Type 1 | –2.37 | 1.0 | 1.0 | 1.2 | 1.1 | 1.1 | 1.5 | -9.3 | -9.5 | 1.27 | –0.40 | 0.86 |
|  | Type 2 | –2.74 | 1.0 | 1.3 | 1.0 | 1.4 | 2.3 | 1.4 | 3.4 | -1.7 | 1.12 | 0.16 | 1.25 |
|  | Type 3 | –3.31 | 1.1 | 1.1 | 1.4 | 1.1 | 1.2 | 1.4 | 0.1 | -5.0 | 1.52 | 0 | 1.53 |





**Table 5:** Contribution rate of urbanization and weather type to ER.

| Source | CR | | |
|---|---|---|---|
| | ER$_{HW}$ | ER$_{ozone}$ | ER$_{total}$ |
| **urbanization** | **27.72%** | **-2.58%** | **8.08%** |
| **Type 1** | **85.36%** | **25.88%** | **46.39%** |
| **Type 2** | **84.51%** | **22.07%** | **43.58%** |
| **Type 3** | **86.54%** | **20.14%** | **45.54%** |
