# Peer review of "Joint Occurrence of Heatwaves and Ozone Pollution and Increased Health Risks in Beijing, China: Role of Synoptic Weather Pattern and Urbanization"

_Atmospheric Chemistry and Physics, 2021_

## Author Comment (AC1)

**Responses to referee(s) comments**

**Dear Editor,**

**Thank you for handling our manuscript. We appreciate the opportunity to receive very pertinent advices from all referees. Their comments are very constructive, and now we have revised our manuscript taking into consideration of all referees' comments. Based on their helpful suggestions, we believe that now we should have appropriately and adequately addressed all the referees' issues and concerns. Please find our point-by-point responses below.**

**Anonymous Referee #1:**

The authors have presented a manuscript that identifies the health effect of heatwaves and ozone in Beijing, China. The authors identified this effect separately and combined. Furthermore, the authors proposed the identification of the effect for urbanization and synoptic systems, with the aid of composites. In general, the article is interesting but needs improvements.

**RESPONSE: Thank you for your valuable time to review this manuscript. We are grateful for your positive feedback on our work. We have carefully read all the constructive comments carefully and we changed the manuscript accordingly. Please find our point-by-point responses below.**

Major issues:

1. The methodology to identify episodes of interest is clear but the explanation about the exposure functions is weak. The authors just cite Liu et al (2021) and Ying et al (2017) for the temperature and ozone parameters. Also, I do not understand why the authors use RR for Liu et al and ER for Ying (lines 120-125). Specifically, the authors state: "every 1°C increase in the daily Tmax above 31.5°C, the largest RR of mortality caused by high temperature in northern China was 1.002". Then, we see Table 3, the Tmax for Urban HW, the value is 36.1, then how the authors obtained 4.76(4,76,4.77)? I thought that It might be (36.1-31.5)*1.002 = 4.6092, but not. Please, clarify and include a similar explanation for ozone.

**RESPONSE: Thanks for your valuable advice. Although we just referred to the exposure function of Liu et al (2021) and Ying et al (2017) in final calculations, we have extensively reviewed the existing literature of high temperature and O$_3$ exposure risk in China, which were supplemented at lines 137–142 as follows:**

*"Previous studies indicated that there were distinctly different magnitudes of human morbidity and mortality caused by high temperature and O$_3$ overexposure over various geographic regions (Huang et al., 2015; Ma et al., 2015; Wang et al., 2020; Yin et al., 2017). For instance, Huang et al. (2015) revealed that for a 1°C increase above the minimum mortality temperature, the daily mortality increased by 1.04% [95% confidence interval (CI): 0.90 to 1.18], 1.25 (95% CI: 0.71 to 1.79), 1.19 (95% CI: 0.79 to 1.58), and 1.38 (95% CI:0.54 to 2.23) in the nationwide, central China, eastern China, and south China, respectively.".*

**ER is the last step for the health risk for both HW and O$_3$ in the final calculation. As Liu et al (2021) pointed out that every 1°C increase in the daily Tmax above 31.5°C, the largest RR of mortality caused by high temperature in northern China was 1.002. During HW periods, Tmax for urban is 36.1°C, so RR=1+(36.1-31.5)\*1.002%=1.046092 , ER= (RR-1)\*100%=(1.046092-1)\*100%=4.6092%. This is a little bit deviation with the 4.76% in Table 2, which is caused by the fact that we calculated the ER of each day and then averaged. It**

**cannot be ignored that, at the lowest mortality temperature of 31.5°C, the relative risk (RR) is 1. For O₃ exposure, a 10-μg m⁻³ increase in MDA8 O₃ was related to an increase in the total daily mortality of 0.39% (95% CI: 0.04%, 0.75%) in northern China during the warm season (Yin et al., 2017). That is, the coefficients of exposure response function (β) between O₃ and total mortality through nonlinear regression is 0.39%. For HW in urban areas, the value of MDA8 O₃ is 197.1, RR=e^(0.39%*197.1/10)=1.0799, ER=(RR-1)*100%=7.99%. We have corrected this paragraph in revised version. Please also see as follow:**

*"Here, we refer to the coefficients of exposure response function (β) for the high temperature as suggested by Liu et al. (2021), while that O₃ concentration as suggested by Yin et al. (2017) in northern China. In detail, Liu et al. (2021) investigated the mortality caused by high temperature in 84 cities in China from 2013 to 2016, and found that for every 1°C increase in the daily $T_{max}$ above 31.5°C, the largest RR of mortality caused by high temperature in northern China was 1.002 (95% CI: 1.001, 1.004). According to Eq. (2), we can deduce that $\beta_{Tmax}$=0.997% (95% CI: 0.996%, 0.999%), note that RR equals to 1 when $T_{max}$ =31.5°C. For O₃ exposure, a 10-μg m⁻³ increase in MDA8 O₃ was related to an increase in the total daily mortality of 0.39% (95% CI: 0.04%, 0.75%) in northern China during the warm season (Yin et al., 2017), that is, $\beta_{Ozone}$=0.39% (95% CI: 0.04%, 0.75%)".*

2. In Figure 5, the authors show the diurnal cycle for some variables claiming that there are significant differences. Do the authors mean statistical significant in the difference, perhaps after applying a test Mann-Withney? Was this test applied to the point values shown in Figure 5? Finally, neglecting the contribution of O3 precursors to explain the difference in O3 during HW events (line 160), I think it is wrong. Even the authors state in the manuscript that during HW events there are more biogenic VOC emissions. Also, the wind speed is higher during HW, which favours the transport of pollution, from rural to urban areas for instance. Actually, a recent paper published in ACP shows the contribution of local and regional emissions to air quality (https://acp.copernicus.org/articles/21/18195/2021/).

**RESPONSE: Thanks for your constructive advice. We add as supplement the Kruskal-Wallis test for the variables of Figure 5, with a significance of the p-values less than 0.001 for each variable in all three cases (Table S2). The Kruskal-Wallis test is a generalization of the Mann-Whitney test and is suitable for multiple groups of independent samples.**

**Table S2: The Kruskal-Wallis test for figure 5 under different weather conditions (HWs, NHWs and precipitation).**

|     | Source | SS | df | MS | F | P |
|-----|--------|-----|-----|-----|-----|-----|
| **T** | Group | 3.45285E+06 | 2 | 1726424 | 145.03 | 3.21143E-32 |
|     | Error | 9.23655E+06 | 531 | 17394.6 | | |
|     | Total | 1.26894E+07 | 533 | | | |
| **RH** | Source | SS | df | MS | F | P |
|     | Group | 1.30882E+06 | 2 | 654407.7 | 73.42 | 1.13823E-16 |
|     | Error | 6.90874E+06 | 459 | 15051.7 | | |
|     | Total | 8.21756E+06 | 461 | | | |
| **HI** | Source | SS | df | MS | F | P |
|     | Group | 1.90775E+06 | 2 | 953874.7 | 107.02 | 5.75597E-24 |
|     | Error | 6.30981E+06 | 459 | 13746.9 | | |

|  | Total | 8.21756E+06 | 461 | | | |
|---|---|---|---|---|---|---|
| | Source | SS | df | MS | F | P |
| **WS** | Group | 544952.7 | 2 | 272476.3 | 30.57 | 2.29868E-07 |
| | Error | 7672602.8 | 459 | 16715.9 | | |
| | Total | 8217555.5 | 461 | | | |
| | Source | SS | df | MS | F | P |
| **O₃** | Group | 1.48979E+06 | 2 | 744893.9 | 82.15 | 1.4506E-18 |
| | Error | 6.94306E+06 | 463 | 14995.8 | | |
| | Total | 8.43285E+06 | 465 | | | |

**Thanks again for pointing out our mistakes. We have rephrased and added evidence and explanation of emissions at lines 177–202 as follows:**

*"In general, the difference in $O_3$ concentration was mainly due to meteorological conditions and the precursors emission paired with photochemical reactions in the boundary layer. We further investigated the diurnal variation for surface air temperature (T), RH, HI, BLH and WS under HW, NHW and precipitation conditions (Figure 5), and these five variables also showed significant differences (passed the Kruskal-Wallis test of 0.001) in the three periods. For HW days, HI raised more by increased air temperature, and although the RH was relative lower, people still suffered from higher apparent temperature than actual air temperature. Under HW conditions, solar radiation reaching the ground heats the atmosphere increasing the near-surface temperature. Warmer air convection promotes atmospheric instability, with increased WS and higher BLH. It is clear that the meteorological variables at daytime were significantly different during HW periods with respect to NHW periods. Similarly, hourly $O_3$ concentrations also showed significantly difference under different meteorological conditions, and reached the peaks in the afternoon on HW days (Figure 5f). In addition, the contribution of local and regional emissions (transport of pollution between urban and rural areas) to air quality at a city scale should be focused (Thunis et al., 2021), which can also induce urban-rural differences. We assumed that the intraseasonal differences in precursor emissions can be ignored, and further compared the diurnal variation differences in $NO_2$ and CO and $O_3$ between different stations (Figure 6). CO and $NO_2$ levels were higher at traffic stations than urban and suburban stations due to enhanced emission from vehicles, and the lowest CO and $NO_2$ levels appeared at rural stations. Generally speaking, high precursor levels are supposed to correspond to high resultant levels, but the lowest $O_3$ levels were found at traffic stations, followed by rural stations, then urban and suburban stations. Since automobile exhaust in the traffic and urban stations also caused heavily NO emission (Colvile et al., 2001), ambient $O_3$ can be titrated by NO via the reaction $NO + O_3 \rightarrow NO_2 + O_2$ (Gao et al., 2020; Murphy et al., 2007; Sillman, 1999), this process in turn led to higher $NO_2$ levels and the loss of $O_3$ in traffic and urban areas. As for rural stations, low pollutant emissions may be the primary reason for the lower $O_3$ levels. Note that although the CO and $NO_2$ emissions were significantly higher at urban stations than suburban stations, there was less difference in $O_3$ concentrations between these two-type stations, which may be due to $O_3$ consumption induced by titration at urban stations, or more biogenic VOCs at suburban stations. This is because that the difference in $O_3$ concentrations between the rural and the suburban stations were the largest in the afternoon, while the difference in CO and $NO_2$ levels were the smallest, indicating that anthropogenic emissions have less impact in suburban areas, coupled*

*with more than half of suburban stations are covered by vegetations leading to more bio-VOCs emissions"*

[Figure]

**Figure 6: The diurnal variation of (a) CO, (b) NO, (c) O₃, under different stations (shading indicates standard deviation, P < 0.001 means pass the significance test).**

Minor issues:

1. Line 77-80, one paragraph of just one sentence. Each paragraph should have at least three sentences, intro, body and conclusion.

**RESPONSE: Thanks for your constructive suggestion. We have rephrased it as follows: *"Ground-level O₃ observation data during summertime (June–August) of 2014–2019 were retrieved from Beijing Municipal Ecological and Environmental Monitoring Center. After quality control, and excluding stations with a missing-values rate for the O₃ hourly concentration of more than 10%, 31 air quality stations [AQSs; including 11 urban stations, 11 suburban stations, three traffic stations (road monitoring stations for traffic air quality), and six rural stations] are ultimately used in this study. In order to better assess the relationship between O₃ pollution and the meteorological variables, we selected 29 automatic weather stations (AWSs) closest to the environmental monitoring stations from the high-density AWS network. Specific geographic location information can be found in Figure 1 and Table 1. Hourly 2-m air temperature, relative humidity (RH), the daily maximum temperature (T_{max}), and 10-m wind speed (WS) of these 29 AWSs were obtained from the National Meteorological Information Center of the China Meteorological Administration, and then heat index (HI) was retrieved as shown in Rothfusz (1990) as Eq. (1):"*.**

2. Line 115, then again, which beta did you use?

**RESPONSE: Thanks for pointing this out. For high temperature, β_{Tmax}=0.997% (95% CI: 0.996%, 0.999%), for O₃, β_{Ozone}=0.39% (95% CI: 0.04%, 0.75%).**

3. Line 123-124, why do the authors use RR for temperature and then ER for ozone?

**RESPONSE: Thanks for your question. We have elaborated on this question in your major issue 1, and have revised.**

4. Figure 2 is not good. Provide a better figure.

**RESPONSE: Thanks for your constructive suggestion. We have replotted Figure 2, and revised a lot. Please see also as follow:**

"*Figure 2 shows the time series of the HW, NHW, O₃ pollution, and precipitation days, and the interannual and intraseasonal variations of HW and O₃ pollution days. For interannual variation, the total days of O₃ pollution in summer was relative stable, while the total days of HW increased slightly. For intraseasonal variation, O₃ pollution was the most serious in June, while the most frequently HW events in July. Obviously, showing that higher O₃ pollution levels (>160 µg m⁻³) were always accompanied by most HW periods (approximately 79.2% of HW days) in Beijing (Figures 2a and 3b), which were mainly in the middle of summer.*"

[Figure]

Figure 2: (a) Time series of weather types, in which the black dots indicate O₃ pollution that occurred on that day. Interannual (b) and intraseasonal (c) variations in summertime O₃ pollution and HW days.

5. Line 151, which test do the authors use?
**RESPONSE: We used the Analysis of Variance test. We have stated it in revised version.**

6. Lines 165-174: I understand that we might expect lower risk in traffic and urban station for ozone, but you mentioned in line 166 that ozone caused a reduction of 2.44% which means risk lower than 1. More explanation is needed.
**RESPONSE: Thanks for your kind suggestion. We have revised the description in this section and explained it in more detail at lines 203–218 as follows:**
"*Moreover, the high temperatures on HW days not only brought a higher public risk related to high-temperature exposure, but also increased mortality related to O₃ exposure. During HW periods, high temperatures and strong solar radiation accelerate the rate of the photochemical reaction that produces O₃ (Pu et al., 2017; Sun et al., 2017), favouring the production and accumulation of O₃, thereby aggravating health risks. Regardless of the type of stations, the O₃ and high-temperature stressful conditions suffered by the human during HW days has greatly increased. Specifically, for all stations, HWs have increased the ER caused by high temperatures and O₃ by 3.867% (90% CI: 3.863%, 3.875%) and 7.9 (90%CI: 0.78%, 15.78%), respectively (Table 2). The high temperature risks were mainly manifested as followings: urban stations >*"

*traffic stations > suburban stations > rural stations, but the health risks aroused by O$_3$ exposure in different underlying surface stations were more difficult to quantifying due to the complexity of O$_3$ photochemical reactions. As mentioned above, urbanization-enhanced NO or CO titration reduced more O$_3$ loss in urban areas, which was more pronounced over traffic stations. For suburban stations, the abundant biogenic VOC emitted by vegetation also contributed to O$_3$ generation, bio-VOC emissions enhanced more especially in hot days (Ma et al., 2019; Trainer et al., 1987; Wang et al., 2021a). As a result, O$_3$ exposure risks in Beijing were mainly characterized by suburban stations > urban stations > rural stations > traffic stations. Urbanization seems to have increased the ER induced by both high temperatures and O$_3$ exposure. In details, summertime HW, O$_3$ and compound ER increased by 1.67%, 0.20%, and 1.89%, respectively, compared to rural stations. Note that urbanization has alleviated O$_3$ pollution to a certain extent, and the health risk of O$_3$ at stations with developed transportation was even lower than that of rural stations."*

7. Line 177, Please, do not overuse abbreviations, WPSH is not needed.
**RESPONSE: Thanks for your advice. We deleted it.**

8. Line 196, UHI is not defined.
**RESPONSE: It have defined at line 64.**

---

## Author Comment (AC2)

**Responses to referee(s) comments**

**Dear Editor,**

**Thank you for handling our manuscript. We appreciate the opportunity to receive very pertinent advices from all referees. Their comments are very constructive, and now we have revised our manuscript taking into consideration of all referees' comments. Based on their helpful suggestions, we believe that now we should have appropriately and adequately addressed all the referees' issues and concerns. Please find our point-by-point responses below.**

**Anonymous Referee #2:**

While the health influences of deteriorated air quality and extreme weather have been assessed extensively, this work took a step forward to assess the lumped effects under the joint occurrence of both contributors and distill the relative significance of synoptic weather and urbanization in these key patterns that we concern. The results provide new information that fills gaps in the understanding of integrated impacts of ozone and heatwave on public health in Beijing. However, several important issues need to be clarified and addressed before its publication.

**RESPONSE: Thank you for your valuable time to review this manuscript. We are grateful for your very positive feedback on our work. Those comments are all valuable and very helpful for revising and improving our paper, as well as the important guiding significance to our researches. We have studied comments carefully and have made correction which we hope meet with approval. Please find our point-by-point responses below.**

Major comments

1. This work would need a professional edit before the final publication. The content is basically understandable but with a significant number of grammatical errors. Some terminology seems to be inappropriate such as "public mortality risk", "compound risk", "urban hyperthermia", "ozone aggravated" etc. All these make it tough to read through in particular the introduction and discussion.

**RESPONSE: Thanks for your helpful advice. We have carefully checked inappropriate wording and made corrections.**

2. Some previous works are not properly referred at the introduction. For example, some cited literature in lines 39, 43, and 46 is not supportive of the corresponding statements.

**RESPONSE: Thanks for your correction. We removed the inappropriate references.**

3. Scientific theories and existing evidence should be referred to and expressed more precisely. For example, in line 59, "39.6% increase in premature mortality" is in against to the annual mortality in Beijing or other value? More statements can be refined for lines 37-38, line 39, and lines 166-167.

**RESPONSE: Thanks for your helpful advice. We have rephrased these statements as follows:**

**"*On the other hand, the increased $O_3$ concentration induced by urbanization was found to translate to a 39.6% increase in premature deaths (Yim et al., 2019).*"**

**"*Meanwhile, the rapid development of urbanization induced many more emission of hydrocarbons and nitrogen oxides into the atmosphere from traffic vehicle and industries, the rising concentrations of these precursors coupled with high temperature and intense solar radiation during HWs can accelerate photochemical reaction rate and generate more $O_3$ (Sillman, 1999; Yim et al., 2019; Zanis et al., 2000).*"**

*"As a result, O₃ exposure risks in Beijing were mainly characterized by suburban stations > urban stations > rural stations > traffic stations. Urbanization seems to have increased the ER induced by both high temperatures and O₃ exposure. In details, summertime HW, O₃ and compound ER increased by 1.67%, 0.20%, and 1.89%, respectively, compared to rural stations. Note that urbanization has alleviated O₃ pollution to a certain extent, and the health risk of O₃ at stations with developed transportation was even lower than that of rural stations."*

4. The introduction is not very organized and the overall logic flow is not fluent. The authors should further polish this section.

**RESPONSE: Thanks for your constructive suggestion. We have reorganized the introduction section as follow:**

*"Climate warming and rapid urbanization have led to increases in the frequency and duration of extreme high-temperature episodes (Lehner et al., 2018; Meehl & Tebaldi, 2004; Wang et al., 2021b; Yang et al., 2017; Li, 2020). Such prolonged extreme high-temperature exposure can induce an increase in the morbidity and mortality due to cardiovascular and respiratory diseases, posing a serious threat to human health (Patz et al., 2005; Xu et al., 2016). Therefore, the extreme high-temperature events are recognized as one of the most serious types of meteorological disaster worldwide. However, high temperatures during summer heatwaves (HWs) are paired with serious O₃ pollution frequently, for instance, significantly increased O₃ concentrations have been observed in the UK and France during the August 2013 heatwave event (Lee et al., 2006; Vautard et al., 2005; Vautard et al., 2007). High concentrations of O₃ exposure would stimulate the human respiratory system, damage lung cells, and aggravate other chronic lung diseases (WHO, 2021), which poses a great threat to human health. Consequently, residents may suffer from dual health risks caused by both high temperatures and O₃ exposures in summer. Although extreme hot events have received extensive attention from academia and society, the research on health risks aroused by O₃ pollution associated with high temperature has been neglected. As a result, it might be greatly underestimated that the health risks to the human body enduringly exposed to the outdoors during hot days.*

*As a continuous extreme case of high temperature weather in summer, heat waves (HWs) have previously been shown by numerous epidemiological studies to cause significantly higher overall deaths than non-heatwave (NHW) periods (Conti et al., 2005; Fouillet et al., 2006). Subsequently, many scholars launched investigations on the relationship between high temperature exposure and mortality (Abbas and Tewtel-Salem, 2005; Huang et al., 2015; Zhang et al., 2017), and they found that when the temperature was higher than a certain threshold temperature, the mortality rate increased with the increase of temperature. Most studies suggested that there were a U-, V-, W-, or J-shaped non-linear change relationships between daily mortality and daily temperature (Goggins et al., 2012; Huang et al., 2015; Y. Zhang et al., 2017). Similar studies on O₃ concentration and mortality have also been progressing (Atkinson et al., 2012; Gu et al., 2018; Pope et al., 2016). Particularly, some epidemiological evidences showed that the coefficient of the O₃ concentration–response relationship for mortality in summer was higher with respect to other seasons (Pattenden et al., 2010; Pope et al., 2016), suggesting that the health effects and mortality related to O₃ pollution were exacerbated by hot temperatures. Therefore, the significant increase in O₃ concentrations during summertime is also greatly responsible for the increase in excess mortality, that is, high temperatures and O₃ exhibit a joint impact on public health (Hertig et al.,*

*2020; Katsouyanni et al., 1993;Pattenden et al., 2010). Numerous previous studies have been devoted to the individual impacts of a single extreme high-temperature or air pollution event on human health (Ma et al., 2015; Ning et al., 2020; Wang et al., 2020; Wong et al., 2013; Xu et al., 2016). However, with the co-occurrence of extreme HW and O₃ pollution events in summer becoming more frequent, it is imperative to reveal the underlying mechanisms of extreme HW–O₃ compound events and to improve the level of risk assessment related to extreme events in urban areas (Sartor et al. 1995; Hertig et al., 2020).*

*Together with the rapid development of economic globalization and urbanization, human activities and the changes in the urban underlying surface have induced frequent occurrences of both extreme high surface air temperature and air pollutions (Chew, et al., 2021; Li et al., 2016; Luo & Lau, 2018, 2019; Meehl et al., 2007; Rastogi, 2020; Wang et al., 2007; Yang et al., 2020; Zheng et al., 2020). Particularly, HWs paired with the urban heat island (UHI) effect exposes urban residents to more sustained extreme high temperatures (Chew et al., 2021; Jiang et al., 2019; Tan et al., 2010; Wang et al., 2017; Zong et al., 2021b). Meanwhile, rapid urbanization induced many more emissions of hydrocarbons and nitrogen oxides into the atmosphere from traffic vehicle and industries, the rising concentrations of these precursors coupled with high temperature and intense solar radiation during HWs can accelerate photochemical reaction rate and generate more O₃ (Sillman, 1999; Yim et al., 2019; Zanis et al., 2000). As a result, urban residents may face more severe stresses from both heat and O₃ pollutions. However, note that the improvement of economic level, medical infrastructure and air-conditioning utilization associated with urbanization can alleviate the health burden of the human body in the face of high temperature and O₃ exposure to a certain extent (Bai et al., 2016; Kovach et al., 2015; Li et al., 2017).Therefore, it can be concluded that there still are some uncertainties in affecting the excess mortality of high temperature and O₃ pollution. To sum up, clarifying the formation mechanism of HW–O₃ compound events and quantifying their health risks to urban residents are important scientific issues that warrant further investigation.*

*Beijing, the capital of China, is the second largest city in the country, with a permanent population of 21.89 million. It is not only one of the fastest developing metropolises in China in recent decades, but also a typical heat island city (Ren et al., 2007; Wang et al., 2017; Yang et al., 2013). Taking Beijing as a typical example, therefore, this study focuses on the health risks of extreme HW–O₃ compound events during summertime of 2014–2019, and comprehensively investigates the roles of synoptic weather patterns and urbanization in these compound events based on surface observation and reanalysis data. Then, the contributions of weather types and urbanization to the excess mortality induced by combined heat and O₃ stress were quantified according to the established health assessment model. The findings are expected to provide a scientific reference for the monitoring and forecasting of summertime HW–O₃ compound events and their health risks from the perspective of synoptic patterns and urbanization in high-density mega cities."*

5. Although involved in the previous work, the weather type classification method should be at least briefly described to improve the integrity of the paper.

**RESPONSE: Thanks for your constructive suggestion. We made some supplements to this section as follow:**

*"The T-mode principal component analysis (T-PCA) is an improved mathematical method to*

*classify the circulation pattern, which has a low dependence on preset parameters, and has advanced temporal and spatial stability of classification (Richman, 1981; Huth et al., 2008). It decomposes the original data matrix into the product of the principle component matrix and the load matrix (two low-dimensional matrices), then rotates the first r (r ≤ n) principal components with larger variance contributions obliquely, and finally obtains the synoptic patterns and classifications of each time according to the magnitude of the load (Huth et al., 2000). Consequently, T-PCA has been widely used in the studies of atmospheric circulation effects of extreme weather (Miao et al., 2019; Ning et al., 2019; Yang et al., 2018, 2021; Zhang and Villarini, 2019; Zong et al., 2021a). Here, T-PCA was applied in COST733class to classify the 850-hPa GH field of the joint occurrence of HW and $O_3$ pollution events and the number of classifications was determined based on the explained cluster variance, more specific details on T-PCA were introduced in our previous study (Zong, et al., 2021a). As for the categorical data, we mainly focused on the domain (110°–125°E, 32°–47°N), including Beijing, associated with these 84 days of compound events during summertime (June–August) 2014–2019.”*

6. The first paragraph in section 3.1 (line 142-160) took efforts to presents the relationship of ozone concentration with different meteorological indicators, which however have been systematically studied already. I recommend simplifying this part.

**RESPONSE: Thanks for your constructive suggestion. We made significant adjustments to Section 3.1, and added evidence and explanation of emissions, see details as below:**

[revised manuscript text omitted]

**RESPONSE: Thanks, changed accordingly.**

2. In Table 3 and Tale 4, what does the red color represents?

**RESPONSE: Red color indicates groups with greater ER, and we added the legend into the table note.**